# MDMP: Multi-modal Diffusion for supervised Motion Predictions with uncertainty

## Abstract

*This paper introduces a Multi-modal Diffusion model for Motion Prediction (MDMP) that integrates and synchronizes skeletal data and textual descriptions of actions to generate refined long-term motion predictions with quantifiable uncertainty. Existing methods for motion forecasting or motion generation rely solely on either prior motions or text prompts, facing limitations with precision or control, particularly over extended durations. The multi-modal nature of our approach enhances the contextual understanding of human motion, while our graph-based transformer framework effectively capture both spatial and temporal motion dynamics. As a result, our model consistently outperforms existing generative techniques in accurately predicting long-term motions. Additionally, by leveraging diffusion models' ability to capture different modes of prediction, we estimate uncertainty, significantly improving spatial awareness in human-robot interactions by incorporating zones of presence with varying confidence levels.*

## 1. Introduction

Through collaboration and assistance, robots could significantly augment human capabilities across diverse sectors, including smart manufacturing, healthcare, agriculture, construction and many others. Indeed, they can complement the critical and adaptive decision-making skills of human workers with higher precision and consistency in repetitive tasks. However, one challenge prohibiting human-robot collaboration is the safety of workers in the presence of robots. To act safely and effectively together, continuous knowledge of future human motion and location in the common workspace with a measure of uncertainty is pivotal. This real-time awareness allows robots to adjust their trajectories to avoid collision and perform precise collaborative tasks [5, 25, 58].

Humans can predict future events based on their self-constructed models of physical and socio-cultural systems.

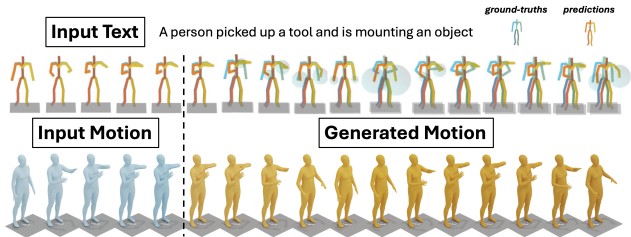

Figure 1. MDMP integrates skeletal motion and text to generate long-term motion predictions with uncertainty zones, shown in both skeletal and 3D human mesh formats.

This skill, developed from childhood through observation and active participation in society, enables them to anticipate others' movements. Researchers are now trying to transfer this capability often refered as "Theory of mind" [10] to machines by training them to learn similar motion estimation tasks. Current methodologies fall into two main categories: Human Motion Forecasting (see Section 2.2) and Human Motion Generation (see Section 2.3). While the former uses only a short input sequence of skeletal motion to predict its future trajectory, the latter relies exclusively on textual prompts to generate motion sequences.

Despite advancements in text-to-motion models, challenges remain in controlling generation due to the expansive action space a simple prompt can describe, which may not always align with human expectations or behavior. Moreover, while some text-to-motion methods have been adapted to perform tasks like motion editing or motion prediction by conditioning their generative process on motion data during sampling, our study demonstrates that, since they are only fed with textual prompts during training, our method consistently outperforms them in terms of accuracy metrics.

Conversely, motion prediction using past sequences is a long-standing challenge that has achieved high accuracy over short-term predictions but struggles with long-term predictions. Even for humans, predicting someone's immediate future movement based on past motions is feasible, but beyond one or two seconds, the multitude of possibilities makes it nearly impossible without context. However, knowledge of the intended action provides a rough idea of

A person puts their hands up while walking and then puts them down.

A man postures his arms like holding a dance partner and dances the waltz.

He grabs something with his right hand and then walks to his left and grabs something with his left hand.

The person was pushed while walking but did not fall and continued.

From a standing position, the person slowly walks in circle, clockwise, and then stops.

Figure 2. **SMPL [34] Meshes of MDMP Predicted motions of different scenarios.** The text descriptions vertically associated to the motions as well as the blue frames are the inputs of the model. The orange frames are the predictions, darker colors indicate later frames.

future positions, as the contextual information of the action guides intuition.

In Human Robot Collaboration (HRC), there is a crucial need for longer-term predictions to coordinate precise interactive tasks, avoid collisions, and maintain efficient trajectory planning. As a result, our method uniquely combines and synchronizes textual and skeletal data to generate precise, longer-term predictions. Indeed, this integration allows for a richer, more contextually aware generation of motion predictions. To the best of our knowledge, this model is the first to be trained on a combination of both types of inputs to leverage context in motion.

In this work, inspired by MDM [50] (Motion Diffusion Model) and LTD [37] (Learning Trajectory Dependencies), we propose a transformer-based diffusion model with a Graph Convolutional Encoder optimized for the spatio-temporal dynamics of motion data. A key design element is the use of learnable graph connectivity, as introduced by Mao et al. [37], to more effectively capture joint dependencies. Additionally, our Multi-modal Diffusion Model for Motion Predictions (MDMP) harnesses the stochastic nature of diffusion models to predict presence zones with varying confidence levels. This uncertainty measure is particularly crucial for long-term motion predictions, where uncertainty grows over time. By offering a spatial understanding of human presence, our model significantly enhances collision avoidance, improving safety and real-time interaction in dynamic collaboration scenarios.

We summarize the contributions as follows: 1) A novel multi-modal diffusion model trained on both textual and skeletal data for precise long-term motion predictions. 2) An uncertainty estimation method to significantly enhance spatial awareness and safety in HRC scenarios. 3) A graph-based transformer capturing spatial-temporal dynamics effectively. 4) A comprehensive validation of uncertainty estimation, with an open-source implementation.

## 2. Related Work

In this section, we review key works that inform our approach. We cover Diffusion Generative Models, Human Motion Forecasting, and Human Motion Generation, highlighting the advancements and limitations in each area as they relate to our method.

### 2.1. Diffusion Generative Models

Diffusion models [18, 46, 47] are neural generative models based on a stochastic diffusion process as modeled in Thermodynamics. The training process involves two phases: forward and backward. The forward process takes observed samples x and progressively adds Gaussian noise until the original information is completely obscured. In contrast, the backward or reverse process employs a neural model that learns to denoise a sample from pure noise back to the original data distribution p(x), hence the term Denoising Diffusion Probabilistic Models [18]. DDPMs have gained prominence in generative modeling, initially demonstrating excellent performance in image generation, and later in conditioned generation [9] and latent text representation [41] using CLIP [44]. Recently, diffusion models have also been applied to various generation tasks, such as text-to-speech [43], text-to-sound [56], and text-to-video [19].

While diffusion models excel in performance, a significant trade-off is the lengthy inference time required for the reverse process, which is impractical for real-time applications. However, many work such as DDIM [48] and Consistency Models [49] tackles that issue and trade off computation for sample quality. Nichol et al. [40] found that instead of fixing variances of distributions modeling the progressively denoised data as a hyperparameter [18], learning it would improve log-likelihood, forcing generative models to capture all data distribution modes, and enable faster sampling with minimal quality loss. Considering the paramount importance of efficiency in HRC, we follow Nichol et al.'s approach by learning variances and leverage the different modes as a factor for uncertainty. Our method demonstrates

better performance with just 50 time steps instead of 1000, achieving over 20 times faster inference.

## 2.2. Human Motion Forecasting

Human Motion Forecasting aims to predict future full-body motion trajectories in 3D space based on past observations from motion capture data or real-time Human Pose Estimation methods. This task is formulated as a sequence-to-sequence problem, using past motion segments to predict future motion. Deep learning methods have shown notable results due to their ability to learn motion patterns and understand spatio-temporal relationships. Early methods employed RNNs [11, 20, 26, 32, 38], then CNNs [31, 57] and GANs [8, 13, 17, 22, 27, 53, 60] but either accumulated errors led to unrealistic predictions or faced limitations due to prefixed kinematic dependencies between body joints. GCNs have proven effective for the task [7, 29, 30, 33, 37, 59], considering that the human skeleton can be effectively modeled as a graph. Transformer-based models, leveraging self-attention [51] for long-range dependencies, have also been adopted [2, 4, 39, 54]. Considering the efficiency and accuracy of the previously mentioned methods, our denoising model leverages GCNs to encode joint features due to their effectiveness in capturing spatial patterns, and a Transformer backbone in the latent space to address the temporal nature of motion data. However, since none of these methods can learn contextual information from the data they are fed, they tend to diverge for durations beyond one second.

## 2.3. Human Motion Generation

Instead of predicting future motion based on past sequences, some generative methods are conditioned on natural language [1, 42] to overcome this short-term issue. This approach faces other challenges such as the vast variability of possible motions corresponding to the same label. However, Text2Motion has garnered significant interest and varied successful approaches. TEMOS [42] and T2M [15] employ a VAE to map text prompts to a latent space distribution of language and motion. MotionGPT [21] furthers this by proposing a unified motion-language framework. MDM [50] proved that diffusion models are a better candidate for human motion generation, as they can retain the formation of the original motion sequence and thus allows them to easily apply more constraints during the denoising process. Then, LDM [6] performed the Diffusion in the latent space and MoMask [16] leveraged Masked Transformers.

By fixing some parts of a motion sequence and filling in the gaps, some of these Text2Motion baselines such as MDM [50], MotionGPT [21] and MoMask [16] propose a form of "motion editing" by forcing their models to generate motions with preserved original data. Unlike these methods, which only edit motions during sampling, our approach trains the model with both textual prompts and motion sequence conditioning to learn contextuality and guide generation towards precise predictions. While these models are compared on diversity and multi-modality metrics, our goal is to minimize the distance between predictions and ground-truth for accurate predictions in HRC.

## 3. Methodology

We now explain the architecture of our proposed MDMP in detail. For an overview, please refer to Figure 3. As part of the Diffusion Process MDMP progressively denoises a motion sample conditioned by the input motion through masking. Our architecture employs a GCN encoder to capture spatial joint features. We encode text prompts using CLIP followed by a linear layer; the textual embedding $c$ and the noise time-step $t$ are projected to the same dimensional latent space by separate feed-forward networks. These features, summed with a sinusoidal positional embedding, are fed into a Transformer encoder-only backbone [51]. The backbone output is projected back to the original motion dimensions via a GCN decoder. Our model is trained both conditionally and unconditionally on text, by randomly masking 10% of the text embeddings. This approach balances diversity and text-fidelity during sampling.

Our method uses the building blocks of MDM [50], but with three key differences: (1) a denoising model that includes variance learning to increase log-likelihood and perform uncertainty estimates, (2) the GCN encoder with learnable graph connectivity, and (3) a learning framework that incorporates contextuality by synchronizing skeletal inputs with initial textual inputs.

### 3.1. Problem Formulation

A motion sample can be represented by a temporal skeleton sequence $X = \{p^i\}_{i=1}^N$ of length $N$ where a frame $p_i$ denotes a pose that can be modeled using different joint feature representations depending on the dataset (see Section 4.1). The simplest form that any representation can easily revert to without any loss of information is the joints' position in 3D space where $p_i = \{x(1)_i, ..., x(J)_i\}$ with joints $x(j)_i \in \mathbb{R}^{M=3}$ and $J$ being the total number of joints. Some parameterizations use rotation matrices ($M = 9$), angle-axis ($M = 4$), or quaternion ($M = 4$) to represent each joint, some also include information such as angular and/or linear velocity.

### 3.2. The Variational Diffusion Process

A Diffusion model can be described as a Markovian Hierarchical Variational Auto-Encoder [35] with a constant latent dimension. During training, we draw $X_0$ from the data distribution, and at each time step $t$, the fixed encoder adds linear Gaussian noise centered around the output of the previous latent sample $X_{t-1}$ until its distribution becomes a stan-

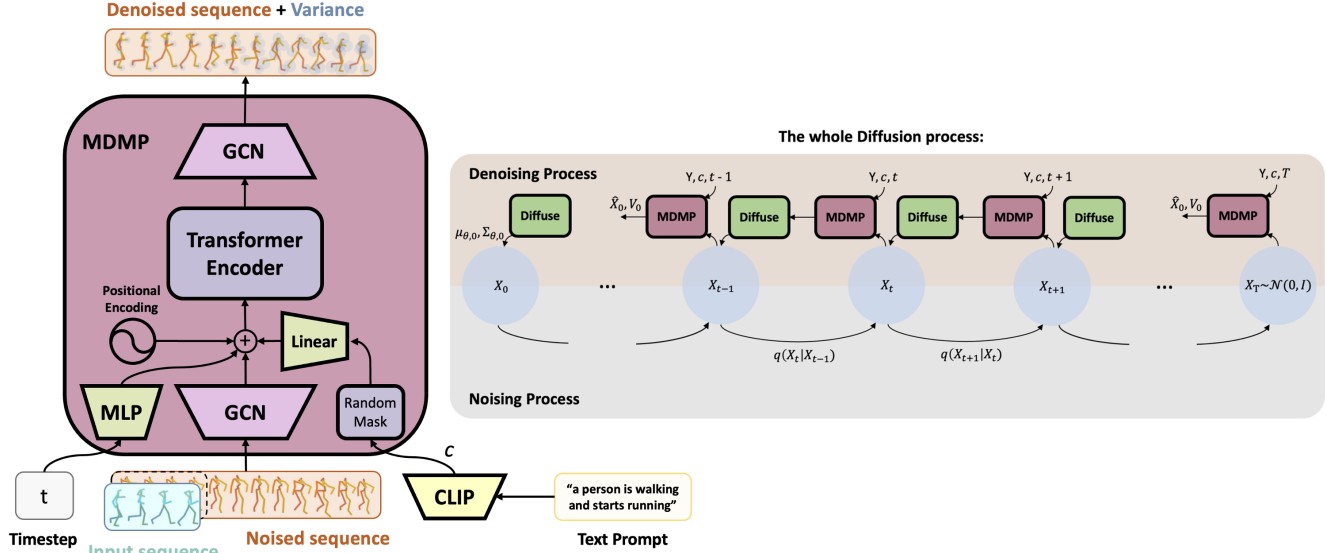

Figure 3. **(Left) Architecture of MDMP.** The denoising model takes as input a motion sample $X_t = \{p_t^i\}_{i=1}^N$ from the previous latent distribution, the diffusion time step $t$ and the conditioning parameters: $Y = \{p^i\}_{i=1}^n$ with $n < N$ the motion input sequence and $c$ the textual embedding encoded by CLIP [44]. At each time step, MDMP outputs a prediction of the final motion $\hat{X}_0$ along with $V_0$, the variance of each predicted joint feature. **(Right) Overview of the Diffusion Process.** On top is the denoising Process, where the Sampling starts from $t = T$ and recursively calls MDMP and uses the output along with $X_t$ to diffuse back to $X_{t-1}$ by calculating $\mu_{\theta,t}$ and $\Sigma_{\theta,t}$.

dard Gaussian at the final time step $T$. Hence, the Gaussian encoder is parameterized with mean $\mu_t(X_t) = \sqrt{\alpha_t}X_{t-1}$ and variance $\Sigma_t(X_t) = (1 - \alpha_t)I$:

$$q(X_t|X_{t-1}) = \mathcal{N}(X_t; \sqrt{\alpha_t}X_{t-1}, (1 - \alpha_t)I) \quad (1)$$

$$q(X_{1:T}|X_0) = \prod_{t=1}^{T} q(X_t|X_{t-1}) \quad (2)$$

Inspired by Nichol et al. [40], we use a cosine scheduler for $\beta_t$ and $\alpha_t = 1 - \beta_t$ such that $\beta_t, \alpha_t \in [0, 1]$. $\alpha_t$ is slowly decreasing, so that for $T = 1000$, $\alpha_t$ is small enough to say that $X_T \sim \mathcal{N}(0, I)$.

Then, during both training and inference, we use MDMP (see Fig 3) as the decoder—conditioned at each step by the previously mentioned inputs $Y$ and $c$—to progressively denoise $X_t$ from a standard Gaussian. Instead of predicting the noise $\epsilon_0$ as formulated in DDPM [18], we follow [45] and [50] and predict the signal itself along with its variance: $\hat{X}_0, V_0 = \text{MDMP}(X_t, t, Y, c)$

Then we use this prediction $\hat{X}_0$ along with the current $X_t$ to diffuse back to the posterior mean:

$$\mu_{\theta,t-1} = \frac{\sqrt{\alpha_t}(1 - \bar{\alpha}_{t-1})X_t + \sqrt{\bar{\alpha}_{t-1}}(1 - \alpha_t)\hat{X}_0}{1 - \bar{\alpha}_t} \quad (3)$$

$$\text{with} \quad \bar{\alpha}_t = \prod_{s=1}^{t} \alpha_s. \quad (4)$$

We use the simple objective from [18] to train our model:

$$L_{\text{simple}} = \mathbb{E}_{X_0 \sim q(X_0|c,Y), t\sim[1,T]} \left[ \|X_0 - \hat{X}_0\|^2 \right] \quad (5)$$

One subtlety is that $L_{\text{simple}}$ provides no learning signal for variances, as Ho et al. [18] chose to fix the variance rather than learn it. However, in our framework, we leverage learned variances to generate presence zones with varying confidence levels to help ensure safety in HRC scenarios.

### 3.3. Learning the Variances of the Denoising process

To learn the reverse process variances, our model outputs a vector $V_0$ of the same shape as $\hat{X}_0$, and-following Nichol et al. [40]—we parameterize the variance as an interpolation between $\beta_t$ and $\tilde{\beta}_t$ in the log domain by turning this output $V_0$ into $\Sigma_{\theta,t}$ as follows:

$$\Sigma_{\theta,t} = \exp(V_0 \log \beta_t + (1 - V_0) \log \tilde{\beta}_t) \quad (6)$$

$$\text{with} \quad \tilde{\beta}_t := \frac{1 - \bar{\alpha}_{t-1}}{1 - \bar{\alpha}_t} \beta_t. \quad (7)$$

Then, we leverage the reparameterization trick $x_t = \bar{\alpha}_t x_0 + \sqrt{1 - \bar{\alpha}_t}\epsilon$ with $\epsilon \sim \mathcal{N}(0, I)$ to sample from an arbitrary step of the forward noising process and estimate the variational lower bound (VLB). As mentioned earlier, the diffusion model can be thought of as a VAE [23] where $q$ represents the encoder and $p_\theta(x_{t-1}|x_t) = \mathcal{N}(x_{t-1}; \mu_{\theta,t}, \Sigma_{\theta,t})$ is the decoder, so we can write:

$$L_{\text{VLB}} := L_0 + L_1 + \ldots + L_{T-1} + L_T \quad (8)$$

$$L_0 := -\log p_\theta(x_0|x_1) \quad (9)$$

$$L_{t-1} := D_{\text{KL}}(q(x_{t-1}|x_t, x_0)\|p_\theta(x_{t-1}|x_t)) \quad (10)$$

$$L_T := D_{\text{KL}}(q(x_T|x_0)\|p(x_T)) \quad (11)$$

Finally, with $q(x_{t-1}|x_t, x_0) = \mathcal{N}(x_{t-1}; \tilde{\mu}(x_t, x_0), \tilde{\beta}_t I)$ we estimate $L_{t-1}$ and approximate $L_{\mathrm{VLB}}$ with the expectation $\mathbb{E}_{t,X_0,\epsilon}[L_{t-1}]$.

Since $L_{\mathrm{simple}}$ does not depend on $\Sigma_{\theta,t}$, we define a new hybrid objective: $L_{\mathrm{hybrid}} = L_{\mathrm{simple}} + \lambda L_{\mathrm{VLB}}$

Conversely to Nichol et al. [40], we apply a clamping on $V_0$ to prevent NaN values during the calculation of $L_{\mathrm{VLB}}$.

### 3.4. Encoding the joint features with GCNs

To encode the spatial pose features, we leverage GCNs [52]. Instead of relying on a predefined sparse graph, we follow Mao et al. [37] and learn the graph connectivity during training, thus essentially learning the dependencies between the different joint trajectories. To this end, we use a fully-connected graph with $N$ nodes, $N$ being the length of the predicted sequence. The strength of the edges in this graph is represented by the weighted adjacency matrix $A \in \mathbb{R}^{N \times N}$. The graph convolutional encoder/decoder then takes as input a matrix $H^{(\mathrm{in})} \in \mathbb{R}^{N \times F}$, where in our case $F$ is the number of body joint features. Given the input a matrix $H^{(\mathrm{in})}$, the adjacency matrix $A$ and a set of trainable weights $W \in \mathbb{R}^{F \times \hat{F}}$, a graph convolutional layer outputs a matrix of the form: $H^{(\mathrm{out})} = AH^{(\mathrm{in})}W$. All operations are differentiable with respect to both the adjacency matrix $A$ and the weight matrix $W$, which allows training on both.

### 3.5. Predicting Uncertainty

To derive an effective uncertainty index for each joint prediction over time, we explore three different approaches which we evaluate and compare in Section 4.4:

- **Mode Divergence:** This approach measures the variability between multiple motion sequences generated from the same input. We compute several predictions in parallel, calculate the standard deviation of these sequences, and use this as the uncertainty index.
- **Denoising Fluctuations:** Here, we measure the fluctuations during the denoising process as an uncertainty indicator. As illustrated in Figure 1 which tracks the evolution of the x-coordinate of key joints (head, hands, feet) from random noise to the final prediction, earlier steps are very noisy and progressively converge with more or less stability. Significant fluctuations in the last 20 timesteps are used as an indicator of uncertainty.
- **Predicted Variance:** The final approach uses the learned variance of the predicted distribution of each motion sequence $\Sigma_{\theta,0}$ as the uncertainty factor.

Both the second and third methods produce outputs in the same dimensions as the model, including predictions for root height, root angular and linear velocity, as well as joint positions and velocities in the local reference of the root. To calculate a single uncertainty index for each joint at each timestep, we average all features associated to the same joint.

## 4. Experiments and Results

In this section, we present the experimental setup and evaluation of our proposed model. We describe the dataset used for training and testing, outlines and explains the choice of metrics used for accuracy and uncertainty, and provide details on our model's implementation. Our comprehensive quantitative and qualitative evaluation, includes comparison with state-of-the-art Text2Motion baselines that propose Motion-Editing re-implemented for a fair comparison with similar conditioning, analysis of uncertainty parameters, and an ablation study to assess the effects of motion-text fusion and architectural design choices.

### 4.1. Dataset

To train and evaluate our model, we use the HumanML3D [15] dataset, which is the largest and most diverse collection of scripted human motions. It combines motion sequences from the HumanAct12 [14] and AMASS [36] datasets, processed to standardize the motions to 20 FPS with a maximum length of 10 seconds per sequence. HumanML3D comprises 14,616 motions with 44,970 descriptions, covering 5,371 distinct words, totaling 28.59 hours of motion data with an average length of 7.1s and three textual descriptions per sequence. The dataset is split for training and evaluation. For evaluation, we filter the set to include only motions longer than 3s, allowing us to condition the models on 2.5s of motion and predict at least 0.5s into the future up until more than 5s for longer recorded motions. After filtering, the evaluation set contains 4,328 out of the initial 4,646 motion sequences.

### 4.2. Metrics for Accuracy and Uncertainty

To evaluate and compare the accuracy of our model we use the *Mean Per Joint Position Error* (*MPJPE*) on 3D joint coordinates, which is the most widely used metric for evaluating 3D pose errors. This metric calculates the average L2-norm across different joints between the prediction and ground truth. Since HumanML3D [15] pose representation contains 263 redundant features per body frames including joint positions, velocities and rotations we use a transformation process (described in the Appendix) to obtain the 3D joint positions in order to both calculate the *MPJPE* and visualize the predicted sequences.

To further validate our method we have also added some more metrics in the C.4 Section of the Appendix. First of all, we have re-trained MDM [50] with skeleton data as an input for a direct comparison, demonstrating the efficacy of our architecture. Secondly, one issue with the *MPJPE* is that it is biased towards one "ground-truth sequence" and thus heavily penalizes frequency or phase shifts common in longer-term predictions, leading to misleadingly large errors even if motions remain qualitatively realistic. Hence we compared our method with baselines on the

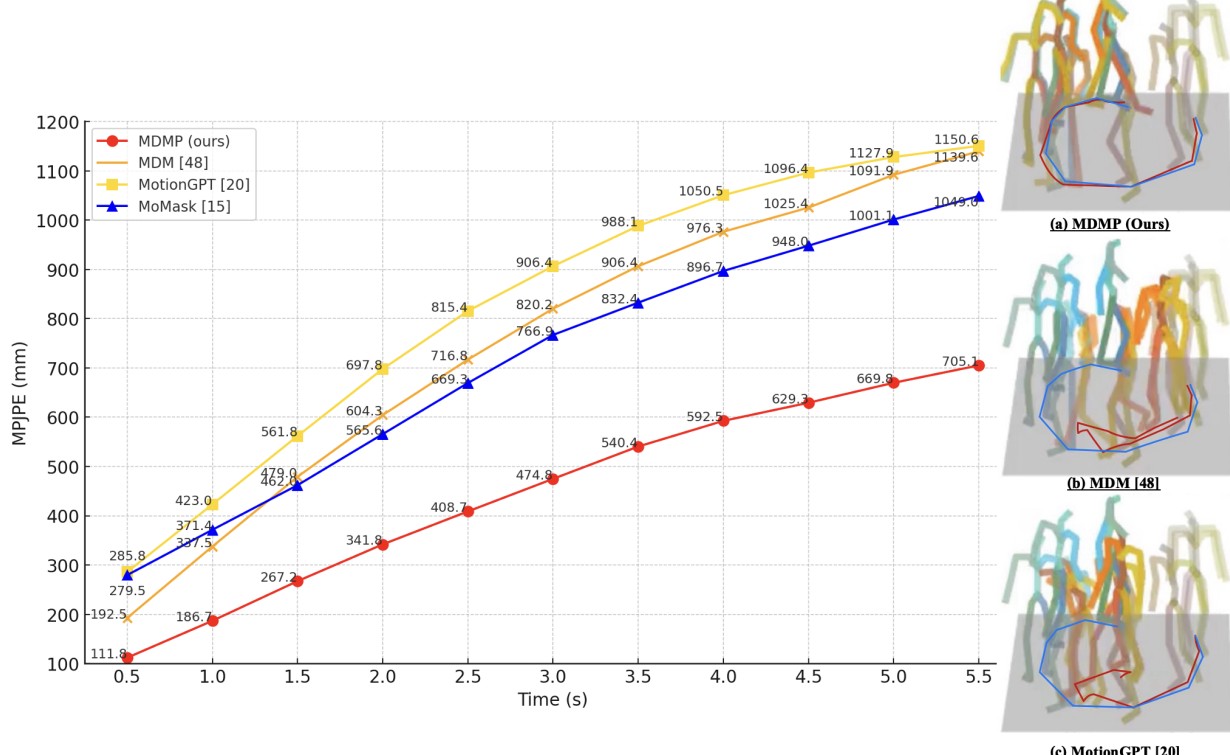

Figure 4. **(Left) Temporal evolution of error in predictions.** Quantitative Results on HumanML3D over *MPJPE* [mm]. **(Right) 3D Plots of Motion Predictions (orange) vs Ground truth (blue).** Motion Sequence example associated to textual prompt:*"from a standing position, the person slowly walks in circle, clockwise, then stops"*. Paler shades represents earlier frames.

NPSS [12] metric which measures similarity in frequency spectra rather than absolute frame-by-frame error, making it better suited to assess the quality of long-term predictions by capturing perceptually relevant motion coherence. Finally, we report results using metrics proposed by Guo et al. [15], such as *Frechet Inception Distance* (*FID*), *R-Precision*, and *Multimodality*. However, these metrics primarily assess motion quality, semantic alignment with textual input, and variability rather than precise spatial accuracy. Additionally, they depend on pretrained feature extractors not tailored to motion-conditioned predictions.

To evaluate and compare our uncertainty indices, we use sparsification plots, a common approach for assessing how well estimated uncertainty aligns with true errors [3, 24, 28, 55]. In our implementation, we compute multiple motion sequences and rank each joint's uncertainty. By progressively removing the joints with the highest uncertainty and summing the remaining error, we obtain the sparsification curve. The ideal reference, known as the "Oracle", is based on ranking joints by their true errors. A well-performing and reliable uncertainty index should produce a curve that decreases monotonically and closely follows the oracle.

### 4.3. Implementation Details

Our models were trained on an *NVIDIA Titan V* GPU over 1.7 days and on *NVIDIA Tesla V100* GPU over 1.2 days with a batch size of 64. We used 8 layers of the Transformer Encoder with 4 multi-head attention for each, separated by a GeLU activation function and a dropout value of 0.1. The GCN layer encodes the joint features from $X_t$ into a latent dimension of 1024 when learning variances and 512 without learning variances. 1024 corresponds to the concatenation of the joint features of $\hat{X}_0$ [512] and $V_0$ [512]. To encode the text, we use a frozen CLIP-ViT-B/32 model. Each model was trained for 600K steps, after which a checkpoint was chosen that minimizes the *MPJPE* metric to be reported. Our generative process is conditioned by a motion input sequence of 50 frames which represents 2.5 seconds at 20FPS. We also set $\lambda = 0.001$ to prevent $L_{\text{VLB}}$ from overwhelming $L_{\text{simple}}$. We evaluate our models with guidance-scale $\mu = 2.5$ but as discussed in the Motion & Text ablation study Section 4.4 this can be adapted for specific applications (eg. short/long-term predictions).

To evaluate the effectiveness of our multimodal fusion approach, we compare against state-of-the-art Motion Editing baselines MoMask [16], MotionGPT [21] and MDM [50] which are all trained on HumanML3D [15].

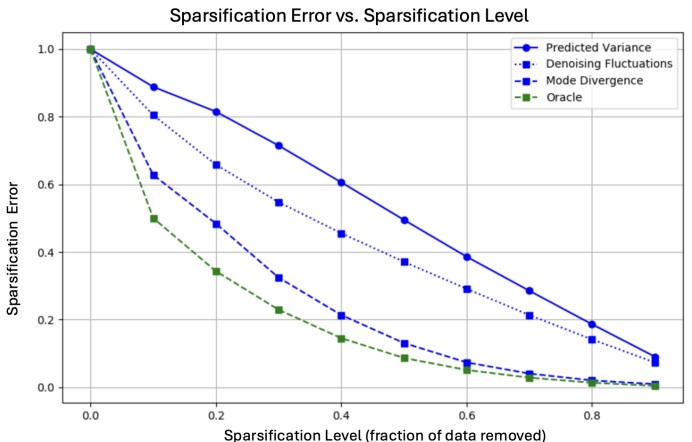
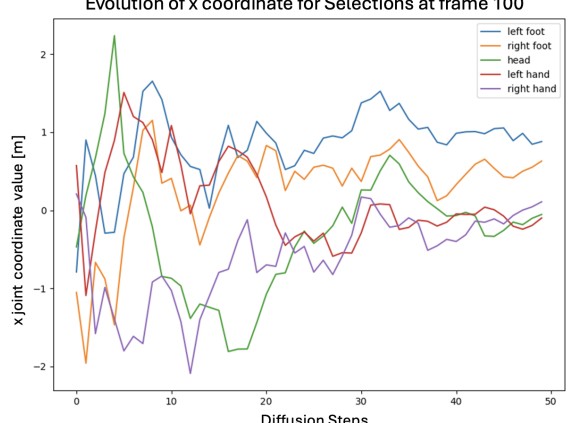

Figure 5. **(Left) Sparsification Error Plot.** Quantitative Results of the uncertainty parameters: The Mode Divergence index closely follows the Oracle curve, indicating the strongest alignment between uncertainty estimates and true errors. **(Right) Joint Position Evolution over the Denoising Process.** The position is progressively denoised until it converges to its final prediction. The fluctuations are used as a parameter for uncertainty.

We implement their pretrained versions (open-sourced) and compare on the entire test set of HumanML3D using *MPJPE*. Conversely to Motion Editing, to ensure a fair comparison setting we conditioned each baseline with only the same motion prequel sequence of 50 frames and compared the rest of the predicted sequence to the ground-truth.

### 4.4. Quantitative & Qualitative Results

**Model Accuracy Evaluation over *MPJPE*:** Unlike the implemented baselines MoMask [16], MotionGPT [21] and MDM [50] that treat motion data as a masked input during sampling, our model is trained to leverage it as an additional supervision signal, which we find to lead to significantly enhances performance, especially over longer sequences. In Fig. 4 the temporal evolution chart shows that our model outperforms these baselines in accuracy, with consistently lower *MPJPE* values over time and a more gradual increase in error. These results are also demonstrated qualitatively in the 3D plots Fig. 4 **(Right)** (see Appendix & Video for more examples) where our predictions align more closely with the ground truth, especially towards the end of the sequence. Indeed, both baselines' outputs fail to follow the indicated trajectory (projection of the root joint in the XZ-plane) whereas our model follows the "circle", almost aligning with the ground truth on the last frame.

**Uncertainty Parameters Evaluation:** The results of our comparison study between the different uncertainty indices are presented in Fig. 5 **(Left)**. The Sparsification Error plot (explained in 4.2) shows that the best-performing index is the Mode Divergence, which closely follows the Oracle curve, indicating a strong alignment between uncertainty estimates and true errors. These results are also demonstrated qualitatively in the video as well as in the Additional Experimental Results (Appendix) where we visualize the

evolution of the zones of presence with varying confidence levels based on the different uncertainty indices. For clarity and visibility, we limit the uncertainty visualization to the "end-effector" joints—specifically the head, hands, and feet—since these are the most critical in human-robot collaboration, and visualizing uncertainty for all joints would create overly cluttered visuals. We calculate the mean uncertainty across the x, y, and z coordinates for each key joint, using this value as the radius of the sphere representing uncertainty around the end-effectors.

**Uncertainty Results Interpretation:** Although the Denoising Fluctuations and Predicted Variance methods show a general decline in their sparsification curves, the effect is less pronounced, suggesting these indices are less reliable for uncertainty estimation. The learned variance is supposed to generally follow the same trends as the original fixed schedule, consistently decreasing during denoising to reduce stochasticity. However, its effectiveness as an uncertainty factor is somewhat limited, as the final value, while still meaningful, becomes slightly less informative. Similarly, the instability of fluctuations diminishes their reliability. In contrast, the Mode Divergence factor consistently rises over time, aligning with the increasing error, making it the most robust and dependable indicator (see video and Appendix for visual confirmation in 3D plots).

**Ablation Study - Motion and Text Effects:** To evaluate the relevance of our multi-modal contribution, we perform an ablation study, presented in Table 1, comparing our standard approach to one where models are fed with either motion or textual inputs exclusively. Firstly, this study clearly confirms that combining both types of inputs results in significantly higher prediction accuracy. Secondly, the study indicates that our model relies more heavily on motion input sequences than textual prompts. Notably, it performs

| Time (seconds) | 0.5s | 1s | 1.5s | 2s | 2.5s | 3s | 3.5s | 4s | 4.5s | 5s | 5.5s |
|---|---|---|---|---|---|---|---|---|---|---|---|
| **Ours with motion & text** | 111.8 | **186.7** | **267.2** | **341.8** | **408.7** | **474.8** | **540.4** | **592.5** | **629.3** | **669.8** | **705.1** |
| **MDM [50] with motion & text** | 192.5 | 337.5 | 479.0 | 604.3 | 716.8 | 820.2 | 906.4 | 976.3 | 1025.4 | 1091.9 | 1139.6 |
| **Ours with text no motion** | 254.8 | 418.6 | 609.5 | 796.8 | 972.2 | 1105.1 | 1253.3 | 1383.8 | 1526.1 | 1624.8 | 1679.8 |
| **MDM [50] with text no motion** | 237.9 | 362.6 | 482.9 | 595.5 | 687.8 | 783.2 | 871.6 | 965.3 | 1039.6 | 1085.2 | 1143.8 |
| **Ours with motion no text** | **100.2** | 186.9 | 271.9 | 358.9 | 445.7 | 528.6 | 608.7 | 677.6 | 739.1 | 810.8 | 902.0 |
| **MDM [50] with motion no text** | 406.1 | 614.5 | 852.3 | 1079.3 | 1288.6 | 1503.5 | 1684.8 | 1871.9 | 2001.6 | 2187.3 | 2332.0 |

Table 1. Ablation study: *MPJPE* (mm) to assess Motion and Text Effects

| Time (seconds) | 0.5s | 1s | 1.5s | 2s | 2.5s | 3s | 3.5s | 4s | 4.5s | 5s | 5.5s |
|---|---|---|---|---|---|---|---|---|---|---|---|
| **Encoder/Decoder: Linear** | 118.6 | 205.5 | 298.8 | 385.5 | 472.0 | 551.3 | 629.7 | 692.0 | 741.0 | 791.9 | 852.1 |
| **Encoder/Decoder: GCN** | **111.8** | **186.7** | **267.2** | **341.8** | **408.7** | **474.8** | **540.4** | **592.5** | **629.3** | **669.8** | **705.1** |
| **Learning the Variance: False** | **86.3** | **163.0** | **250.1** | **332.4** | 409.3 | 485.4 | 560.5 | 622.6 | 676.8 | 729.5 | 775.5 |
| **Learning the Variance: True** | 111.8 | 186.7 | 267.2 | 341.8 | **408.7** | **474.8** | **540.4** | **592.5** | **629.3** | **669.8** | **705.1** |
| **Diffusion Steps: 1000** | **104.8** | 192.2 | 280.6 | 360.9 | 438.3 | 482.1 | 553.6 | 617.2 | 653.5 | 702.3 | 745.8 |
| **Diffusion Steps: 50** | 111.8 | **186.7** | **267.2** | **341.8** | **408.7** | **474.8** | **540.4** | **592.5** | **629.3** | **669.8** | **705.1** |

Table 2. Ablation study: *MPJPE* (mm) to evaluate Architectural Design and Parameter Choice

slightly better without text for very short-term predictions. This means that our model could be used in a HRC setting for continuous operation between different actions, even without specific action context. This capability is presumably not possible with Text2Motion models, which perform poorly without text, as the study shows. Finally, the study confirms that textual information is most useful for longer-term predictions where the stochasticity and variability of potential scenarios are much higher.

**Ablation Study - Architectural Design and Parameter Choice:** To assess our architectural contributions, we conduct a deeper analysis with additional ablation studies presented in Table 2. In the first study, we retrain our model with both the encoder and decoder composed of simple linear layers, as in MDM [50]. The study confirms that learnable graph connectivity improves the understanding of human joint trajectory dependencies, especially for long-term predictions. The second study evaluates our architectural design that learns the variance of the motion sample distribution. Although learning variances allow diffusion models to capture more data distribution modes with we leverage for uncertainty estimates, our study shows that it only enhance accuracy over long-term predictions. In the final study, inspired by Nichol et al. [40], we significantly reduce the number of diffusion steps from 1000 to 50 which considerably improves the computational efficiency-pivotal for real-time Human-Robot Collaboration-and resulted in improved accuracy over time.

# 5. Conclusion and Limitations

We present MDMP, a multimodal diffusion model that learns contextuality from synchronized tracked motion sequences and associated textual prompts, enabling it to predict human motion over significantly longer terms than its predecessors. Our model not only generates accurate long-term predictions but also provides uncertainty estimates, enhancing our predictions with presence zones of varying confidence levels. This uncertainty analysis was validated through a study, demonstrating the model's capability to offer spatial awareness, which is crucial for enhancing safety in dynamic human-robot collaboration. Our method demonstrates superior results over extended durations with adapted computational time, making it well-suited for ensuring safety in Human-Robot collaborative workspaces.

A limitation of this work is the reliance on textual descriptions of actions, which can be a burden for real-time Human-Robot Collaboration, as not every action is scripted in advance. Currently, we use CLIP to embed these textual descriptions into guidance vectors for our model. An interesting future direction is to replace these descriptions with images or videos captured in real time within the robotics workspace. Since current Human Motion Forecasting methods already rely on human motion tracking data, often obtained using RGB/RGB-D cameras, the necessary material is typically already present in the workspace. Given that CLIP leverages a shared multimodal latent space between text and images, this approach could provide similar guidance while being far less restrictive, making it more practical for dynamic and unsupervised HRC environments.

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
