# OpenReview forum: "MDMP: Multi-modal Diffusion for supervised Motion Predictions with uncertainty"
_thecvf.com/CVPR/2025/Workshop/HuMoGen — CVPR 2025 Workshop HuMoGen Submission_

### Official Review · Reviewer_dqVW · 2025-03-26
**Clear improvements in long-term motion prediction using Denoising Diffusion Probabilistic Models (DDPM) with Graph Convolution Network (GCN) using both text and audio as an input.**

**Rating:** 5
**Confidence:** 3

**Review:**

Summary:
The paper investigates a mixture of Motin Diffusion Models [50] (without additional losses) and Learning Trajectory Dependencies [37] (GCN part of it) and extends it to motion prediction when using both text and some "prompt/past" motion as input. Further, it introduces uncertainty estimation in order to improve spatial awareness of the generated motion. Similar to [37], learning the adjacency matrix of the GCN improves the interjoint dependencies, and further passing using transformer blocks models this information, resulting in clear improvements.

Quality, clarity, and originality:
The paper is well-written and clear, all experiment details are mentioned very well, and it is fairly original.

Pros:
- Strong experimentation and ablations. However, (optionally) I think Table 1 in the appendix should be moved to the main article as the NPSS results should be highlighted as well.
- The experiment demonstrates the advantage of using this approach as MDMP follows the best path compared to other baselines.
- Detailed ablations clearly demonstrate the advantage individual components bring.

Cons/Question:
-  The mode diversity uncertainty measure does not require the DDPM formulation with learnt variance, what is the difference one observes in the sparcification of mode diversity when using the L_simple DDPM formulation instead?
- In supplement, it is stated that:

> We argue that as described by Guo et al. [2], the pretrained motion feature extractor used for computing FID was trained using a contrastive loss to produce geometrically close feature vectors for matched text-motion pairs. Hence, the motion encoder is specifically optimized for motions conditioned solely on text descriptions and may not accurately capture the features of motions generated by models conditioned on both text and an initial motion segment (motion prequel).

However, the result of one of the ablation also shows, and the authors state:

> textual information is most useful for longer-term predictions where the stochasticity and variability of potential scenarios are much higher.

This argument contradicts each other since MDMP generates better long-term predictions compared to other baselines for which text is an important factor, and thus, it should have better FID scores. More clarity on this or better rephrasing/explanation would be nice to have.
- Most if not all the other baselines are text2motion or motion generation baselines. Please also compare with some other motion prediction/continuation baselines if possible.
- While mode divergence is quantified, the advantage of uncertainty estimation or learning the variance for modeling is not evaluated for spatial awareness despite being mentioned as a contribution.

Overall:
- The paper is very well written with strong experimentations and ablations. I believe it would be a great addition to the workshop and should be accepted.

---

### Official Review · Reviewer_ayFD · 2025-03-28
**Review for Multimodal Human Motion Prediction with Uncertainty Modeling**

**Rating:** 5
**Confidence:** 4

**Review:**

## Summary

The paper aims to improve human motion prediction by leveraging a multi-modal approach, where both prior motion and textual motion descriptions are used as conditioning information in a motion diffusion-based framework. Key contributions of the paper are:
1. Including learnable graph connectivity in motion diffusion for capturing spatial-temporal dependencies in human motion.
2. Using variance estimation in the reverse diffusion process to quantify uncertainty on predictions.

The paper follows Mao [2] in capturing the spatial and temporal relations in human motion sequences using Graph Convolutional Networks (GCNs). Specifically, it represents motion trajectories as graphs over time steps, where each node contains the full pose at a given time. The GCN then learns temporal dependencies between frames via a fully connected, learnable adjacency matrix.

The proposed framework models uncertainty by learning the variance of the reverse diffusion process, following Nichol [3], in contrast to fixing it as in the original DDPM approach. This learned variance serves as an explicit uncertainty estimator within the generative process. In addition, the authors propose and evaluate post hoc uncertainty quantification strategies to assess prediction variability. This uncertainty estimation capability is particularly beneficial for human-robot collaboration (HRC) tasks.

To quantify prediction uncertainty, three uncertainty indices are introduced: mode divergence, denoising fluctuation, and predicted variance, which serve as evaluation tools to assess the reliability of motion predictions. Among them, mode divergence shows the strongest alignment with actual prediction error over time, as confirmed through sparsification plots.

Furthermore, the authors provide extensive evaluation on motion prediction error (mean per joint position error – MPJPE). Results show that the proposed system consistently outperforms the evaluated baselines. For a fairer comparison, one of the baselines (MDM) is re-trained to have similar motion context input. The performance gain of the system still holds in this case too, proving the strength of the proposed system.

---

## Strengths

- Novel framework leveraging variance estimation for uncertainty modeling in motion prediction.
- Extensive experiments showing superior performance over baselines, particularly for long-term prediction.
- Well-defined and evaluated uncertainty indicators.
- Relevant for safe human-robot collaboration, a real-world use case.

---

## Weaknesses

- The difference between the motion prediction and motion generation task could raise concerns about fairness, but this is accounted for in the additional re-training studies.
- For the intended use in HRC systems, the dataset used to train this model may be limiting. It would be interesting to see how well the model generalizes to out-of-distribution motions or ambiguous textual inputs, which are common in real-world collaborative scenarios.
- The utility of text input for real-time robotic systems may be challenging, as noted by the authors.

---

## Remarks

- In Section 3.2, the loss derivation comes from Nichol [3], but in its current form it is a bit confusing. Motivating the reintroduction of the L_{VLB} for variance prediction before line 284 would help readability.
- The term “presence zone” is used throughout the text but not properly introduced.
- Figure 5, fluctuation plot: It’s unclear whether the figure shows a single trajectory or variability across multiple diffusion runs. Including confidence intervals or standard deviation bands would clarify the magnitude of denoising fluctuation and better support its use as an uncertainty metric

---

## Small remarks

- **Line 038**: Theory of mind is a bit broader than motion prediction; rephrasing might set the scope better.
- **Line 090**: This statement lacks experimental validation.
- **Line 251**: Variable \( Y \) is used without introduction. Introduce in Section 3.1 for readability.
- **Line 299**: GCN [1] is misattributed—it cites GAT, not Kipf & Welling.
- **Line 308**: Clarify if \( F = J \times M \).
- **Line 326**: Figure 5 is referenced as Figure 1.
- **Line 561**: *MotionClip* [4] could be interesting.

---

## References

[1] Kipf, Thomas N., and Max Welling. *"Semi-supervised classification with graph convolutional networks."* arXiv preprint arXiv:1609.02907 (2016).
[2] Mao, Wei, et al. *"Learning trajectory dependencies for human motion prediction."* Proceedings of the IEEE/CVF International Conference on Computer Vision. 2019.
[3] Nichol, Alexander Quinn, and Prafulla Dhariwal. *"Improved denoising diffusion probabilistic models."* International Conference on Machine Learning. PMLR, 2021.
[4] Tevet, Guy, et al. *"Motionclip: Exposing human motion generation to clip space."* European Conference on Computer Vision. Cham: Springer Nature Switzerland, 2022.

---

### Meta-Review · Area_Chair_QdxF · 2025-03-31

**Recommendation:** Accept

**Metareview:**

This paper presents a multi-modal approach to human motion prediction combining prior motion and textual descriptions within a motion diffusion framework, with learnable graph connectivity and variance estimation for uncertainty quantification.

Strengths
---------
- Novel uncertainty modeling framework for motion prediction
- Consistent performance improvements over baselines, even when fairly re-trained
- Well-defined and evaluated uncertainty indices
- Clear applications for human-robot collaboration
- Thorough experimental validation

Weaknesses
----------
- Questions about generalization to out-of-distribution motions
- Most baselines are text-to-motion rather than motion prediction models
- Potential contradiction regarding textual information's role in long-term predictions vs. FID evaluation
- Spatial awareness benefits of uncertainty estimation need clearer evaluation

Recommendation
--------------
Both reviewers rated this paper for strong acceptance. Based on the reviews, I recommend acceptance with the suggestion to clarify loss derivation, introduce undefined terms, improve Figure 5 with confidence intervals, address the potential contradiction between FID evaluation and text importance, and compare with additional motion prediction baselines. The paper makes valuable contributions that would strengthen the venue.

---

### Decision · Program_Chairs · 2025-03-31

Accept